# Media content analysis of general practitioners' reactions to care.data expressed in the media: what lessons can be learned for future NHS data-sharing initiatives?

Elizabeth Ford ,[1] Yalda Kazempour,[1] Maxwell J F Cooper,[1] Srinivasa Vittal Katikireddi ,[2] Andy Boyd[3]

[1]Primary Care and Public Health, Brighton & Sussex Medical School, Brighton, UK
[2]MRC/CSO Social & Public Health Sciences Unit, University of Glasgow School of Life Sciences, Glasgow, UK
[3]School of Social and Community Medicine, University of Bristol, Bristol, UK

**Correspondence to**
Dr Elizabeth Ford;
e.m.ford@bsms.ac.uk

## ABSTRACT

**Objectives** Care.data was a 2013 UK government initiative to extract patient data from general practices in England to form a centralised whole-population database for service planning and health research. After a public outcry, the scheme was postponed and cancelled. Public views of care.data have previously been analysed; this study aimed to understand contemporary general practitioners' (GPs) views of the scheme, which may have been influential in its downfall.

**Design** Systematic search of media articles, followed by media content analysis.

**Setting** UK-based mainstream and GP-facing media in 2013 and 2014.

**Participants** Articles were eligible if they focused on care.data, and GPs were quoted, authored the article, or if articles were written for a majority GP audience.

**Interventions** N/A.

**Primary and secondary outcome measures** Themes which explained GPs' reactions to care.data and which could explain support for or opposition to the scheme.

**Results** 162 media articles met inclusion criteria and were drawn from newspapers, news websites and GP-facing websites. GPs recognised care.data's potential value for research and improving care, but had grave concerns about the scheme's implementation. These centred the lack of safeguards and purpose around the scheme which meant patients were not able to make informed decisions about opt-out. GPs perceived they were poorly resourced to meet competing demands to both share patients' data and protect confidentiality. They distrusted the government's likely uses of the data and perceived a risk of patient reidentification if the data were sold onto commercial entities.

**Conclusions** Findings show specific concerns which GPs had about care.data which led to the withdrawal of support. Future NHS patient data-sharing schemes should engage with GPs and other clinicians as key stakeholders from the earliest moments of planning, so that their views and needs are incorporated into the design of such schemes.

### Strengths and limitations of this study

► We systematically and comprehensively searched mainstream and GP-facing media published during the unfolding care.data events, to access the voice of doctors as events unfolded, thus minimising recall bias in accounts of events.

► By following a systematic approach, we minimised bias in the selection of articles for the study. However, other sources of GP opinions (such as social media discussions) escaped our analysis.

► Doctors contributing to mainstream news articles may not be representative of GPs in general, but instead can be viewed as key contributors to the prominent circulating discourses at the time.

► GP's views expressed at the time of the debate may not represent their current views on the care.data scheme, its subsequent failure, or on future schemes with similar aims.

## INTRODUCTION
### Care.data
In 2012, laws were passed within the United Kingdom Health and Social Care Act (HSCA) to require all general practitioners (GPs; family doctors) in England to upload their electronic patient records (EPRs) to a central database which could then be linked with additional National Health Service (NHS) datasets and used for research, audit and planning health services. This programme of data collation and reuse was called care.data and was due to be initiated in Spring 2014.[1] It expanded on previous data collections which were limited to hospital records or were opt-in at the GP practice level.[2–5] For care.data and previous schemes, provisions were made so that individual patients who did not wish their data to be used for quality improvement and research purposes could opt out.[1]

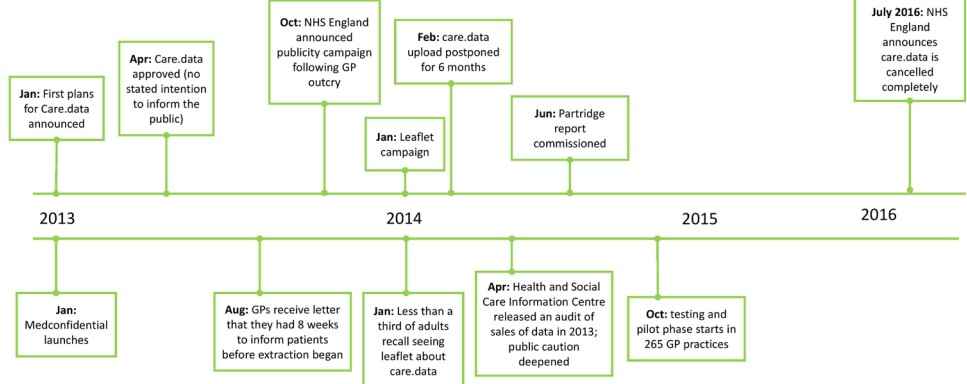

**Figure 1** Summary of key events regarding unfolding of care.data throughout 2013–2014 (based on Presser et al[11]). GPs, general practitioners.

The care.data scheme was advertised to patients in the form of a leaflet 'Better information means better care'[6] which was purportedly delivered to all households in England by Royal Mail in January 2014, although few people reported receiving it.[7] There was no opt-out form on the leaflet and no clear instructions on how to register an opt-out from the scheme.[8] A survey of 1400 members of public showed that only 23% remembered receiving the leaflets and 45% did not understand the care.data scheme.[9] Furthermore, a survey of 600 GPs showed that 80% of them also felt that they did not have adequate knowledge of the scheme.[9]

In early 2014 the programme faced critical reception from patients and other stakeholders, which was widely reported in the media. This adverse media coverage, and the rise of campaign groups,[10] prompted a series of damaging disclosures by the NHS relating to past sharing of centralised health records for purposes beyond those relating to improving the public good.[11] The high media profile and widespread criticism of care.data led to a review of the scheme (the Partridge review[12]), and the UK government postponed the care.data programme in February 2014, finally cancelling it in 2016.[13] A full time-line is shown in figure 1.

Following a lengthy consultation,[14] a new national strategy for an opt-out model for reuse of NHS patient data was rolled out in May 2018 as the 'National Data Opt Out'.[15] At the current time, future universal data-sharing schemes for primary care records under this new strategy are not clear. Given the high likelihood of similar schemes being launched in the near future, it is imperative that as many lessons as possible are learnt from care.data to promote transparency of such schemes and inclusivity of all stakeholders.[16]

### Stakeholder reviews on why care.data went wrong have focused on patient and public views

A body of literature has built up which has looked at themes in the public's objections to care.data and why it became so unpopular. These studies have included analyses of citizens' comments on the care.data website and the Guardian newspaper comments section,[17] and public views posted on Twitter.[8 18] Commentators have drawn lessons from public

reactions to the care.data scheme to suggest recommendations for future schemes,[11] to suggest a theoretical framework for explaining public concern about the scheme[19] and to guide interpretation of new data protection legislation.[20]

### Why focus on GPs?

Any patient data-sharing scheme requires the support of clinicians as central stakeholders in the project, and as the creators and custodians of patient data. In the UK, GPs tend to be the legal owners of the medical information recorded in their EPRs, and in turn are responsible for the way in which these data are processed and used. These responsibilities are enshrined in UK Data Protection Act (the Data Protection Act 1998 at the time of the care.data programme) and the common law duty of confidentiality, and are regulated by the Information Commissioner's Office.

While the British Medical Association (BMA) was initially in favour of the care.data scheme,[21] doctors later wrote publicly about their concerns about the scheme, both before[22] and after it was postponed.[23 24] It is not clear if opposition to the scheme from GPs was one of the main drivers in the postponement and final failure of the scheme, or if GPs' reservations followed the patient, public and media outcry. To understand more about GPs' views on care.data, and what factors underpinned their later opposition to the scheme, we conducted a media content analysis of contemporaneous GP reactions to the scheme expressed in the media. The aims of this study were to systematically and comprehensively identify, and qualitatively analyse, newspaper and news website coverage of the care.data debate within the UK, focussing specifically on articles which quoted GPs or other doctors, or were authored by GPs or other doctors or were written for a solely GP audience. We aimed to describe themes and learning points in doctors' views of the scheme and to look for turning points in opinion during the unfolding events.

### METHODS
### Selection of sources

A systematic search strategy, following principles outlined in Preferred Reporting Items for Systematic Reviews and

**Table 1** Inclusion and exclusion criteria for articles

| Inclusion criteria | Exclusion criteria |
|---|---|
| Article focused mainly (>50%) on the care.data project | Less than 50% of the content related to care.data; |
| AND article quoted a GP or doctor expressing an opinion of care.data; | OR they did not include any quotes or discussion about GPs' or doctors' views; |
| OR was authored by a GP or any doctor | OR the article was a short lead-in to a main story elsewhere in the same edition (which also appears in the sample); |
| OR was written for a majority GP audience | OR it was a summary of, or link to, formal documentation and surveys; |
| OR discussed the opinions, attitudes or views of doctors about care.data | OR it was not a news article in its own right. |

GP, general practitioner.

Meta-Analyses reporting,[25] was used to identify online media about care.data, which quoted, was written by, or was directed towards UK GPs and other medical doctors during the care.data era. Comments posted by readers in response to articles were not used for analysis.

Mainstream newspapers were searched using the database Nexis. This is a comprehensive newspaper database, updated daily, providing full-text access to all UK national newspapers, plus regional newspapers, international news providers and a number of trade journals and magazines. Newspapers were included if they had a circulation in print of 100 000 per day or more. The BBC news website was also searched. Through consultation with GPs (n=4), we identified and searched four GP-facing news websites: Pulse Today, GP online, Royal College of General Practitioner's (RCGP) website and BMA website. The search was constrained to 1 January 2013 to 31 December 2014.

The search terms used in the Nexis database were: 'care.data' AND ('doctor' OR 'dr' OR 'general practitioner' OR 'GP' OR 'clinician' OR 'physician' OR 'specialist' OR 'professor').

The search term used in BBC news and on the GP-facing news sites was care.data.

### Inclusion and exclusion criteria

All articles were screened in full by author YK to identify if they met eligibility criteria (table 1). For any ambiguity in whether articles met inclusion criteria, authors EF and YK worked together to reach a consensus on inclusion. See online supplementary file 1 for full screening results.

### Data analysis

We approached this analysis with a realist ontology and an objectivist epistemology. A thematic analysis was conducted according to the principles of Braun and Clarke[26] in an iterative process.

The content of all articles was imported to NVivo V.12 software. NVivo V.12 software facilitated reading articles and coding themes as they emerged. [26] The first step was for author YK to familiarise herself with the data collected by reading the articles multiple times and noting down initial ideas. Next step was to determine key themes and code features of the data that were relevant to the research

questions. Every article obtained from our search was read and coded line by line to generate themes. Every quote authored by a GP was examined for relevance, and if relevant, was coded. Two authors (YK and EF) discussed initial themes, and consolidated emerging and disparate themes into a smaller number of nested themes. The articles were then read again and all quotes linked to a final list of themes. The articles related to each code were then grouped together under the same theme. Given the large volume of data, themes are represented using indicative quotes that were the most illustrative for that theme, with attention paid to discordant data.

## RESULTS

### Search results

The search returned 923 articles from the Nexis newspaper database (newspapers and web-based publications), BMA website, Pulse Today and BBC, and after screening against eligibility criteria, 162 were found to meet the inclusion criteria. No relevant articles were returned from the search of 'GPonline' or from the RCGP website. The spread of articles is shown in table 2.

### Article characteristics

Characteristics of the articles included for the analysis are summarised in table 3 and full information given in online supplementary file 2.

### Thematic analysis

Thematic analysis revealed five themes regarding GPs' views on care.data. These were:
1. Support for the scheme: care.data would give value for research and may improve care.
2. Concerns about patients lacking informed choice through proposed opt-out system.
3. Concerns about legal responsibilities regarding patient data.
4. Concerns about key safeguards—GPs were unable to reassure patients that they would not be reidentified and their data would not be sold onwards.
5. Concerns about trust between patients and doctors.

**Table 2**  Search results

| Database | No of articles returned by search | No of articles included |
|---|---|---|
| Nexis newspaper print articles | 451 | 83 |
| Nexis web-based articles | 164 | 2 |
| BMA | 48 | 16 |
| BBC news | 24 | 8 |
| Pulse Today | 236 | 53 |
| Total | 923 | 162 |

BMA, British Medical Association.

Numbers given following quotes presented below can be cross-referenced with articles described in online supplementary file 2.

### Support for the scheme: care.data would give value for research and may improve care

Early in the timeline, many GPs were positive about the potential benefits of care.data. The scheme was supported by the RCGP with Dr Clare Gerada, former president of the College, expressing support and committing to work with the BMA and NHS England on the care.data project (5, BMA, May 2013). Dr David Davies, a GP and the Health and Social Care Information Centre's (HSCIC) executive medical director, believed use of linked data would help improve services and allow targeted interventions, by determining patients at higher risks of a particular disease (1, Pulse Today, Mar 2013). Dr Davies explained:

> We'll be linking data, doing risk profiles on the data then feeding it back to GPs to adapt their care. That'll be OK because we're feeding data back to the people who gave it to us for it to be used in direct patient care. (1, Pulse Today, Mar 2013)

Professor Liam Smeeth, a practising GP and professor in London highlighted the necessity of using patients' data to run the NHS :

> These are data that are absolutely necessary to keep the NHS up with the times, to develop new inventions, or to be safe and effective, so we know what's going on in the health service. …There are people worried about confidentiality but I am hoping there aren't people sitting around saying we don't need to do this because we don't need this data. (32, Pulse Today, Jan 2014)

Dr Francesca Lasman, a GP in Cambridgeshire believed the care.data scheme would lead to comprehensive and meaningful analysis leading to insights on managing comorbidities in general practice. She said:

> I think it is vital to develop an understanding of how best to manage complex co-morbidities which exist in general practice. Drug companies and studies focus on one problem but real data from people with multiple conditions and on many medications gives at least a chance of some meaningful analysis and a start for the best approach to tackling prevention and treatments. (39, Pulse Today, Jan 2014)

A number of medical charities expressed support for the care.data scheme in line with NHS England's public awareness campaign. Doctors working for these charities warned the public that if too many people opted out of the scheme, it would impact on science which could lead to increased numbers of deaths in the population (32, Pulse Today, Jan 2014). Professor Peter Weissberg of the British Heart Foundation explained how lack of data could lead to delayed discoveries and more deaths:

> We can carry on making [discoveries] without [it], but we will make them extraordinarily slowly and people will die in the interim. Not only will they die as

**Table 3**  Sources of the articles

| Source | No of articles | No of different authors | No of doctors quotes |
|---|---|---|---|
| The Guardian (including the Observer) | 8 | 8 | 16 |
| The Times | 14 | 6 | 13 |
| The Telegraph (including Daily Telegraph) | 22 | 8 | 29 |
| Daily Mail (including Mail Online) | 22 | 8 | 22 |
| The Independent | 12 | 5 | 14 |
| London Evening Standard | 5 | Unknown | 9 |
| Sky news | 2 | Unknown | 2 |
| Pulse Today | 53 | 11 | 73 |
| BMA | 16 | Unknown | 12 |
| BBC | 8 | 2 | 8 |

BMA, British Medical Association.

a consequence of habits and behaviours they have, which we haven't understood properly, [but] they will die as a consequence of well-meaning attempts by doctors… [who are] treating patients on a daily basis on what we call the evidence base. (32, Pulse Today, Jan 2014)

This generalised support for the scheme was not without qualification, however. The BMA greatly emphasised the importance of gaining public trust and confidence in the scheme. Dr Chaand Nagpaul, the chair of the BMA said:

The BMA continues to support the principle of using anonymised data to plan and improve the quality of NHS care for patients. However, this must be done with the support and consent of the public, therefore patients must be made aware of what is happening to their personal information, what the proposals mean and what their rights are if they do not wish their data to be extracted. (72, Pulse Today, Jan 2014)

### Concerns about patients lacking informed choice through the proposed opt-out system

Doctors were concerned about the care.data opt-out mechanism, and that patients were not well informed enough to make a choice. In August 2013, GPs were given the responsibility of informing their patients of their medical data extraction for care.data during the 8 weeks before the extraction started. However, GPs were not provided extra funding to carry out this task, leading to frustration (154, Pulse Today, Aug 2014). Dr Grant Ingrams, a practising GP in Warwickshire and deputy chair of the General Practitioners Committee's (GPC) IT subcommittee, told Pulse Today:

…I don't think any GP is going to pay hundreds or thousands out of their own pocket to run a campaign. (154, Pulse Today, Aug 2014)

A survey of 427 GPs carried out by Pulse Today and reported on 26 February 2014, revealed that only 19% agreed with the opt-out system with 75% supporting a change to an opt-in system. Ninety per cent of GPs stated that if an opt-out system was used, NHS England should provide an opt-out form with the 'Better Information Means Better Care' leaflets. The GPs expressed concerns that having to opt out could be challenging for particular groups within society, for example migrants (111, Pulse Today, Feb 2014).

RCGP honorary secretary, Professor Nigel Mathers was outspoken about the public's lack of awareness of the scheme and of their right to opt out. He criticised the NHS awareness campaign by stating:

Crucially, where a scheme is based on an opt-out approach, such as in the case of care.data, we believe that it is vital that the NHS is able to show that it is beyond reproach in having done everything practically possible to ensure that patients and the public know

about their right to opt out prior to it going ahead. (82, The Telegraph, Feb 2014)

Professor Mathers also criticised the leaflet scheme, and announced that since it had not worked, all other possible means of communication such as radio, national TV and online adverts should be used. He recommended that personalised letters should be sent to all households in England. Professor Mathers said:

At present, we are concerned that levels of awareness concerning care.data are very low, and believe that there is a strong case for substantial additional activity over and above that already in place to tackle this. (79, Mail Online, Feb 2014)

Following the postponement of the scheme, GPs were surveyed on their opinions. Some GPs believed patients' data should only be extracted with their explicit consent and an opt-in system was the only right way it could be done (143, Pulse Today, May 2014). Some GPs went further and decided to opt out all their patients unless they specifically requested to opt in (141, Pulse Today, May 2014). An Oxfordshire GP who had been practising medicine for 40 years had decided to opt out his entire practice due to lack of information about the scheme. Dr Gancz felt '…*people are being bulldozed into giving consent by default*' (53, The Telegraph, Feb 2014).

A number of GPs who had been in favour of the care.data project initially, later stated that they did not approve of the way the government blamed the lack of public support on GPs. A GP from West Sussex, Dr Jeremy Luke, said:

… [The government] has, as usual, tried to blame GPs for their own failure to engage with and listen to the views of patients. (153, Pulse Today, Aug 2014)

In contrast, some GPs supported an opt-out system and believed an opt-in system would be a retrograde step in research. Dorset GP L-J Evans explained that an opt-in system would rely on every patient consenting for their medical data to be used, which did not work in her opinion. She explained that a disadvantage of the opt-in system would be skewed results and less robust research (150, BMA, Jun 2014).

### Concerns about legal responsibilities regarding patient data

GPs were required to share their patients' data for care.data by the 2012 HSCA which put them in a legally challenging situation, as this sharing might have breached the common law duty of confidentiality. Doctors' duty of confidence regarding the information they hold about a patient can only be lifted where the patient has an expectation of a secondary usage,[27] in this case, to be shared to a centralised NHS data warehouse. GPs perceived that, legally, they could therefore only share data if they were sure that patients were aware of, and fully understood, the care.data scheme and their rights to opt out (111, Pulse Today, Feb 2014). One response to this concern was that some GPs opted out their entire registered list of patients, even though this led them to

be in breach of contract with the 2012 HSCA. A number of doctors who opted out their entire practice believed that in allowing patients' data to be extracted without patients being fully informed, the doctors would not be doing their duty to protect patients' data and would be in breach of the Data Protection Act (19, Pulse Today, Nov 2013).

The GPC deputy chair warned GPs that opting out entire practices would be 'breaking the law', as GPs had a 'statutory duty' according to the Health and Social Care Act to share their patients' data (21, Pulse Today, Nov 2013). However, GPs believed that if patients complained about not being aware of their data being shared, GPs would be held accountable (as they were data controllers) and not the government (33, Pulse Today, Jan 2014). Dr Grant Ingrams, who was a former chair of the GPC's IT subcommittee and a GP in Warwickshire, explained how he thought GPs were 'confounded' between these conflicting Acts:

> …Practices have a lawful obligation under the Health and Social Care Act to send the data to the HSCIC. But an obligation under the Data Protection Act to protect patients' data. It's leaving practices confounded between two rights. If practices aren't sued by one, they'll be sued by another. (6, Pulse Today, Aug 2013)

Later on, a survey indicated that 8% of GPs would opt out all their patients, despite being aware they would face a legal challenge and the possibility of their contract being terminated (111, Pulse Today, Feb 2014).

### Concerns about key safeguards—GPs were unable to reassure patients that they would not be re-identified and their data would not be sold onwards

From the launch of care.data, there had been issues about a lack of transparency and lack of assurance about the confidentiality of data held by HSCIC (139, BMA, Apr 2014; 150, BMA, Jun 2014). One of doctors' main concerns was the possibility of patients being reidentified from their pseudonymised records, this was expressed before the project was even launched (9, Pulse Today, Sep 2013). Data extracted would contain information about the patients' prescriptions, investigation results and some diagnostic codes. However, more sensitive codes such as In-vitro fertilisation (IVF), abuse, HIV, termination of pregnancy, convictions or sexually transmitted infections (STIs) would not be extracted from surgeries. Patient identifiers such as name and date of birth would not be extracted but patients' unique NHS identification number would be. GPs identified that there was a small risk of identification for patients with rare diseases (2, Pulse Today, Apr 2013).

NHS England stated that the data extracted would be crucial to integrate care and reduce treatment and care inequalities. Despite guarantees made by the government, doctors were not reassured (2, Pulse Today, Apr 2013). A GP working in Warwickshire, Dr Paul Thornton stated his concern very early in the conception of the scheme:

> …lots of people will want [the data, including] other Government bodies, the Department for Work and

Pensions, financial institutions, pharmaceutical companies. (2, Pulse Today, Apr 2013).

A survey carried out by Pulse Today revealed that approximately 40% of GPs wished to opt out of the system themselves, as they believed they could potentially be reidentified. An anonymous respondent stated they 'cannot see any clinical justification for the identifiable data extraction' (23, Pulse Today, Dec 2013).

Dr Marie-Louise Tidmarsh, a GP in Derbyshire, worried about being identified by her neighbours and as a result decided to opt out of the system. She said:

> I think patients have been misled about the "confidential" nature of the data extractions, and it is not clear to whom the data may be sold. (39, Pulse Today, Jan 2014)

Moreover, worries about the risk of a substantial database like care.data being hacked was expressed by some doctors (54, Daily Mail, Feb 2014). This risk of hacking or reidentification was perceived to increase with the fact that unknown, and possibly profit-motivated, end-users would be provided with the data. GPs were concerned about the likelihood of the data being sold to private companies (51, Daily Mail, Feb 2014).

Professor Sir Brian Jarman, a 'world specialist on hospital data', claimed that insurance companies could identify patients in less than 2 hours. He explained that tracking down individuals and cross-referencing information could be fairly easy as many of these companies are already in possession of patients' names and home addresses. Sir Brian said:

> I've spoken to analysts who say they can match individual people within a couple of hours. Many organisations, such as insurance companies, hold details of people that include not only name but also postcode, date of birth and gender, which would make it possible to identify named individuals in postcodes and thus have access to their confidential medical information in the care.data database. There is simply too much data and the risks that something leaks are too great. (70, Mail Online, Feb 2014)

A GP in Kent, Dr Ian Williams, indicated he did not believe proposed measures would mask his identity in the data:

> I wish to opt out as I am concerned about identifiable data being moved around and passed on to third parties. I would have no objection if the data held on my records was "pseudonymised" before being extracted from the practice records. (39, Pulse Today, Jan 2014)

### Concerns about trust between patients and doctors

GPs believed that historically, patients have trusted doctors to look after their personal data and make sure of their safety (71, London Evening Standard, Feb 2014). The doctors' relationship with their patients had always been based on trust and patients did not expect any information they share

during consultations to end up in the government's database without their knowledge (25, The Independent, Jan 2014).

The fear of losing their patients' trust was one of GPs' main concerns about care.data. The BMA representatives explained that if patients were not assured of the security of their medical data, it would lead to mistrust and lack of communication by patients. This would mean doctors may not have all the information they need about the patient in the future, and patients would not share all the symptoms that are required to reach a correct diagnosis (112, BMA, Feb 2014; 100, The Daily Telegraph, Feb 2014). The Chair of the Family Doctor Association, Dr Peter Swinyard, explained:

> We have a very basic principle that whatever our patients tell us is confidential. We are completely hamstrung if patients feel they can't tell us something that - in many cases - they wouldn't tell anyone else. It is this threat to total confidentiality at consultations that gets up most GPs' noses. (37, The Independent, Jan 2014)

Professor Sue White of Birmingham University stated this problem could affect particularly vulnerable groups within society, for example women who are experiencing domestic violence issues. She believed that these women are already reluctant to open up to their GPs about their difficulties and if they doubted their data was completely confidential it might make them avoid visiting their GPs (22, The Observer, Nov 2013).

Nested within this theme was the additional distrust between doctors and the government. Because GPs did not trust the government to use patients' data safely, they could not endorse the scheme and this compounded their fear of losing their patients' trust. Dr Philip Bolitho-Jones, a GP partner in Hertfordshire, stated he did not trust the politicians to decide what happened to his or his patients' medical data.

> We don't trust the politicians; they've got no idea what they're doing. They've already leaked: before the system was even set up they were leaking data left right and centre. I wouldn't trust them with my notes; I wouldn't trust them with any of our patients' notes. (141, Pulse Today, May 2014).

## DISCUSSION

Our results showed that GPs supported the care.data scheme in theory at the outset, and agreed that data extraction and reuse was necessary or beneficial for good research, for improving care and for saving lives. However, as time went on, they increasingly opposed certain aspects of implementation of the programme and articulated a number of key concerns. GPs cared about their patients being properly informed about the scheme. They felt the government had not informed patients adequately and could not be trusted to keep patients' data safe. They perceived that the onus was being put on them to make sure their patients were well informed, but no financial or time resources were given to them to achieve this. They disliked being scapegoated for the project's failures in communication. They also perceived that the lack of communication and guidance left them conflicted by competing obligations to share data while maintaining their duty of confidentiality. GPs were fearful that they could end up being held accountable under one or other of these duties.

GPs articulated a conflict over the consent mechanism in the care.data scheme. They felt that an opt-out system would be challenging for some vulnerable groups (eg, migrants). Other literature has identified that this challenge may extend to other marginalised groups such as trans-people.[28] However, GPs understood that an opt-in system would introduce bias, and potentially reduce the quality of research. They also expressed distrust in the system proposed for protecting patient identities within the scheme, this distrust was particularly focused on the government's history of mishandling sensitive records, and the possibility of them being sold to private companies such as insurance companies. A substantial proportion of GPs indicated they did not want their own data to be extracted, because they believed patients could be quite quickly reidentified by private companies with a profit motive. Finally, GPs feared that if patients felt their data were not confidential, patients would not tell them important medical issues, and this would impede GPs' ability to provide good clinical care, and risk harming the most vulnerable patient groups.

Many of the themes expressed by GPs at the time of the care.data scheme tally with those expressed by other stakeholders, about this scheme in particular, and about patient data sharing in general. Citizens writing comments on the care.data and Guardian websites, and publishing microblogs on Twitter, complained of a lack of transparency and poor communication about the scheme, a lack of respect for confidentiality, misgivings about the opt-out mechanism, and fears of erosion of trust in GPs and the NHS.[8 17] They were also concerned about commercial uses of data. Citizens could see the potential benefits of the scheme and felt that if the societal benefits and 'common good' that could be derived from the scheme were made a central tenet of the scheme,[17] and meaningful public and patient involvement could be achieved,[8] the scheme would have been more likely to have been accepted. Reviews of patient opinions on a range of data-sharing schemes[29 30] have also identified similar themes underlying lay thinking on the topic. The majority of respondents to surveys and qualitative studies indicate high but qualified support of sharing patient data for research and other secondary purposes. Participants in these studies articulated fears that their privacy may be compromised and expressed distrust of particular organisations, particular those with a profit motive, which may have access to the data.

Our study additionally exposed some GP-specific issues around the implementation of care.data. GPs felt under-resourced to meet their competing duties under

the HSCA 2012 and the DPA 1998. To meet their duties under these two acts, they needed to make sure patients were fully informed, but were unable to do this without any additional resource coming from the government. They also feared that data-sharing schemes would detrimentally affect their relationships with their patients, leading to an erosion in trust, and leading to them being less able to care for their patients effectively.

### Strengths and limitations of this study

Our approach of analysing contemporary GPs comments on care.data in published newspaper and online news articles made sure we accessed the voice of doctors during events of the unfolding crisis, and not with the hindsight of knowledge of how the situation has evolved since. The search strategy used was comprehensive and designed to include all the mainstream articles written by, quoting or written for doctors about care.data. By following a systematic approach, we minimised bias in the selection of articles for the study. However, it is possible that other sources of news, read by or authored by GPs, were not captured here, or that different opinions were being widely shared through informal GP networks and escaped our analysis. GPs may also have been quoted out of context in the media to suit the purpose of the article. Additionally, those doctors contributing to mainstream news articles may not be representative of GPs in general, but instead can be viewed as key contributors to the prominent circulating discourses at the time. We acknowledge that the discourses of the quoted GPs, which were expressed in 2013–2014, may not represent their current views on the care.data scheme, its subsequent failure, or on future schemes with similar aims.

The thematic analysis was conducted by one researcher and checked carefully by a second, which may make the results more subjective and less reproducible than if a team approach had been taken. However, the themes found in our study map closely onto key themes found in other work in this field,[8 11 17 19] suggesting the results reported are at least consistent with stakeholder views from other sources.

### CONCLUSIONS AND RECOMMENDATIONS

The study suggests that, should a similar programme be introduced in future, the support of GPs should be obtained before implementation in order to improve the success of the programme. GPs' concerns regarding confidentiality, patient information, informed consent, patient trust, need for resources and conflicting legal obligations should be addressed during the planning phase of any data-sharing scheme, and reflected in the implementation of the programme. It appears, from subsequent national initiatives such as the development of the national data opt out,[15] that some of these lessons have already been learnt. During the development of the national data opt out, the opinions of key stakeholders and medical bodies, including doctors, were considered during the

planning phase. There was effective communication with the RCGP and the Royal Colleges of Nursing and Midwifery to ensure the NHS workforce was ready across the country to start the scheme. We recommend going even further, and inviting key primary care stakeholders such as GP representatives, to sit within the teams that design and implement such schemes. Beyond consultation, we recommend a codesign approach which includes stakeholder representatives as equal team members from the outset. Examples of such good practice should be studied further, and can be combined with the results of our study, to ensure a social licence for patient data sharing is achieved with patient and clinicians, before the start of any future scheme.

**Contributors** EF, AB and SVK conceived of the presented idea. YK performed the searches, screening, data extraction and analysis with supervision from EF. MJFC aided interpretation of the results. EF and YK drafted the manuscript. All authors discussed results, contributed to the manuscript and approved the final version.

**Funding** SVK acknowledges funding from a NRS Senior Clinical Fellowship (SCAF/15/02), the Medical Research Council (MRC) (MC_UU_12017/13 & MC_UU_12017/15) and the Scottish Government Chief Scientist Office (SPHSU13 & SPHSU15). AB acknowledges funding from the Avon Longitudinal Study of Parents and Children: MRC and Wellcome (Grant ref: 102215/2/13/2) and the CLOSER longitudinal research consortium: Economic and Social Research Council (ES/K000357/).

**Competing interests** None declared.

**Patient and public involvement** Patients and/or the public were not involved in the design, or conduct, or reporting or dissemination plans of this research.

**Patient consent for publication** Not required.

**Provenance and peer review** Not commissioned; externally peer reviewed.

**Data availability statement** This study used publically available data from main stream media. These data are already in the public domain.

**ORCID iDs**
Elizabeth Ford http://orcid.org/0000-0001-5613-8509
Srinivasa Vittal Katikireddi http://orcid.org/0000-0001-6593-9092

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
