## [Reviewer comments · BMJ Open]

ARTICLE DETAILS

TITLE (PROVISIONAL)	Media content analysis of general practitioners' reactions to care.data expressed in the media: what lessons can be learned for future NHS data sharing initiatives?
AUTHORS	Ford, Elizabeth; Kazempour, Yalda; Cooper, Maxwell; Katikireddi, Srinivasa; Boyd, Andy

VERSION 1 – REVIEW

REVIEWER	Nadine Pohontsch University Medical Center Hamburg-Eppendorf, Germany
REVIEW RETURNED	12-Apr-2020

GENERAL COMMENTS	this was a very interesting article to read, especially for a reader not familiar with the care.data debate. I have some comments/wishes for the manuscript: Could you state the nature of the study (for example, 'qualitative document analysis') in the title of the study? It appears to me that you might not be aware of the COREQ (Tong et al., 2007), as some information is missing in the article and you did not upload the completed checklist. Not every item is applicable to your article, but many are. I especially miss information about the researcher characteristics (Item 1-5). This would help the reader to judge on potential influences on data analysis. Also, what methodological orientation/epistemology underpins your study (Item 9). Could you add information on that? The description of the analysis used is rather short, readers unfamiliar with qualitative research might want to know more about the different steps of the methods used (from familiarizing with the material to themebuilding), and the reasons why you chose thematic analysis to work with the material. I am not familiar with Neis (and many non-English readers might also not know so much about this database), so I would love to get some more information about it. It seems to me, that in line 444 (...work in this field) references to prove that statement are missing. In reference 19 appears to be missing some information too. All in all, your article was an interesting read about what happens when you do not inform and involve the people affected in political processes.
--

REVIEWER	Dr Elka Humphrys Visiting Researcher at the University of Cambridge, UK
REVIEW RETURNED	16-Apr-2020

GENERAL COMMENTS	Research question/study objective The study was interesting to read and the analysis reflected the aim
--

	of the study as stated in the abstract: “to understand contemporary general practitioners’ views of the scheme, which may have been influential in its downfall.” However, on page 6 (lines 122-124) the authors state that the study was conducted to explore the following gap in understanding: “It is not clear if opposition to the scheme from GPs was one of the main drivers in the postponement and final failure of the scheme, or if GPs’ reservations followed the patient, public and media outcry.” Based on the study data it is not possible to assess whether GP opposition was a driver in the scheme failing. It is also not possible to determine if the GP opposition occurred after patient/public outcry as the GP data has not been compared to patient/public media data. It would be beneficial to clarify the purpose of the study. Conclusion The wording of the conclusion comes across as contradictory in that the authors appear to make recommendations for improving future initiatives, such as taking a co-design approach, despite the fact that the National Data Opt-Out scheme has already used this approach. Therefore although the conclusions reflect the results of the study, the recommendations seem slightly redundant when contextualised with current initiatives. Limitations of the study A limitation that should be considered is that GPs may have been quoted out of context in the media to suit the purpose of the article. Please see the attached file for minor comments relating to data clarification and formatting – Please contact publisher for this file.
--	--

VERSION 1 – AUTHOR RESPONSE

Reviewer	Comment	Response	Page and Line Number
1	Dear authors, this was a very interesting article to read, especially for a reader not familiar with the care.data debate.	Many thanks for your positive and constructive review, we found your comments and suggestions very helpful.	-
	Could you state the nature of the study (for example, 'qualitative document analysis') in the title of the study?	We would term this study as a media content analysis. We have added this to the title.	P1, L1
	It appears to me that you might not be aware of the COREQ (Tong et al., 2007), as some information is missing in the article and you did not upload the completed checklist. Not every item is applicable to your article, but many are. I especially miss information about the researcher characteristics (Item 1-5). This would help the reader to judge on potential influences on data analysis.	Thank you for this suggestion which will help to raise the standard of our reporting. We looked at the COREQ but found so many of the items were not relevant. Instead we decided to complete the ENTREQ checklist: Enhancing transparency in reporting the synthesis of qualitative research: the ENTREQ statement. We have completed and uploaded the checklist.	Supplied separately

	Also, what methodological orientation/epistemology underpins your study (Item 9). Could you add information on that?	We have added the following statement in Methods: We approached this analysis with a realist ontology and an objectivist epistemology.	P7, L156
	The description of the analysis used is rather short, readers unfamiliar with qualitative research might want to know more about the different steps of the methods used (from familizing with the material to themebuilding),and the reasons why you chose thematic analysis to work with the material.	We have added further detail about the step-by-step approach we took to analysis.	P7, lines 164-176
	I am not familiar with Neis (and many non-English readers might also not know so much about this database), so I would love to get some more information about it.	We have added a sentence explaining what Nexis is.	P6 lines 139-141
	It seems to me, that in line 444 (...work in this field) references to prove that statement are missing. In reference 19 appears to be missing some information too.	We have added four references here to indicate that we mean the same literature we referenced in the introduction. Reference 19 which is the Carter et al 2015 one, seems complete to us, so we have not changed it.	P17 L465
2	The study was interesting to read and the analysis reflected the aim of the study as stated in the abstract: "to understand contemporary general practitioners' views of the scheme, which may have been influential in its downfall."	Thank you	-
	However, on page 6 (lines 122-124) the authors state that the study was conducted to explore the following gap in understanding: "It is not clear if opposition to the scheme from GPs was one of the main drivers in the postponement and final failure of the scheme, or if GPs' reservations followed the patient, public and media outcry." Based on the study data it is not possible to assess whether GP opposition was a driver in the scheme failing. It is also not possible to determine if the GP opposition occurred after patient/public outcry as the GP data has not been compared to patient/public media data. It would be beneficial to clarify the purpose of the study.	Thank you for pointing this out. We still feel that the stated gap in our understanding is important to draw attention to, even if we did not find the answer to it in our study. Therefore we have left this statement in but decoupled it from the aim of the study. We have stated instead "To understand more about GPs' views on care.data, and what factors underpinned their later opposition to the scheme, we conducted a media content analysis of contemporaneous GP reactions to the scheme expressed in the media. "	P5 L124-125

	The wording of the conclusion comes across as contradictory in that the authors appear to make recommendations for improving future initiatives, such as taking a co-design approach, despite the fact that the National Data Opt-Out scheme has already used this approach. Therefore although the conclusions reflect the results of the study, the recommendations seem slightly redundant when contextualised with current initiatives.	We have reworded this paragraph to demonstrate that many of our suggestions have already been actioned in the national data opt-out scheme but that beyond consultation of stakeholders when designing a scheme, planning could go further and include stakeholder representatives as full team members to enable co-design.	P17 lines 472-482
	A limitation that should be considered is that GPs may have been quoted out of context in the media to suit the purpose of the article.	We have added this to the limitations section.	P17 L456-457
	Page 5, line 91 – the ‘review of the scheme’ needs to be clearly identified as the Partridge report so that the manuscript text clearly maps on to the timeline in Figure 1. Currently the introduction of the Partridge report in Figure 1 is confusing.	We have named the Partridge review in this sentence.	P4 line 91
	Page 5, line 92 – it would be useful to state the month that the government postponed the programme (February 2014) as this information is relevant to contextualise the data presented on page 12.	We have added February to this sentence.	P4 line 92
	Page 5, lines 94-95 – “drawn from Presser” gives the impression that Figure 1 is a direct replica of Presser’s model, rather than an altered version based on Presser’s depiction of the timeline. Using the term ‘based on’ (or similar) rather than ‘drawn from’ would provide clarity. As the model has been adapted for the purposes of the current study, it would be beneficial to extend the timeline to 2016 to depict when the care.data programme was cancelled. This would be the full timeline as per the manuscript text (page 5, lines 92-93).	We have changed this to "based on" in the caption. We have extended the figure to include the 2016 cancellation of the care.data programme.	P4 line 94-95
	Figure 1 – needs a footnote to give the full name of HSCIC. What is the October 2015 testing and pilot phase in 265 GP practices? This needs a brief description in the manuscript.	We decided to change the figure to give the full name of the HSCIC.	Figure

	Page 7, line 133 – If the principles of PRISMA reporting are followed, a PRISMA diagram should be used to give the details of included articles and to document the number of excluded articles along with the reasons for exclusion (based on individual exclusion criteria). The latter information is currently missing from the manuscript for the 761 excluded articles.	We have added PRISMA flowcharts for the sifting and searching of all articles from all sources in Supplementary Materials 1. Unfortunately, we did not record the number of articles rejected for all the exclusion criteria, these were: less than 50% of the article was about care.data (by word count); no discussion of GP views or quotes, article was short lead in to a main story or a summary or link to formal documents, guidelines or surveys, or was not a news article in its own right (e.g. was an advert).	Supp Materials 1
	Table 3 – Daily Mail articles n=21, however based on Appendix 1 there were 22 Daily Mail articles. There were also two Sky News articles that have not been documented in Table 3.	Appendix 1 is now Supp. Materials 2. We have checked the appendix against the table and have updated the table accordingly (the appendix was correct).	P8 line 191
	Page 11, line 239 – when did Pulse carry out the survey of 427 GPs?	We have added in the date of the report of the survey - 26 Feb 2014.	P10 line 258
	Page 16, line 396 – need to distinguish between vulnerable groups identified in the analysis (i.e. migrants, source 111) and vulnerable groups identified by other literature (i.e. trans-people, reference 28).	Thank you for picking this up, we have changed the sentence to indicate that the two exemplar marginalised groups were identified from different sources.	P15 line 415-416
	Page 3, line 41 – “...sold onto to commercial...” – remove ‘to’.	Removed	P2 L41
	Page 7, line 140 – full name for RCGP required as this is the first time this acronym is used in the text.	Added	P6 L144
	Page 7, Table 1 – Second inclusion criteria, ‘And’ should be capitalised (AND).	Changed	P6 L153
	Page 8, Table 2 – Pulse should be ‘Pulse Today’ to be consistent with the rest of the article.	Changed and checked throughout article.	Throughout
	Page 9, line 189 – full name for HSCIC required as this is the first time this acronym is used in the text.	Added	P9 L207
	Page 9, line 165 – Full names for RCGP and BMA may not be needed here if the acronyms have been previously defined (RCGP on page 7, BMA on page 6).	We have removed the full names.	P7 Line 181 and 183

	Page 9, Thematic Analysis: Where a quote is used directly after an explanation of the source, it is not strictly necessary to give the quote in the explanation and at the end of the quote i.e. line 195-196: "Professor Liam Smeeth, a practising GP and professor in London highlighted the necessity of using patients' data to run the NHS (32, Pulse Today, Jan 2014): ...[quote]...(32, Pulse Today, Jan 2014).	We have removed these duplicate citations where they directly precede the quote.	Throughout results
	Page 11, line 229: Theme title does not quite match the stated theme title on page 9, line 177 – page 11 says 'with', page 9 said 'through'.	These have been reconciled.	P10 line 248
	Page 11, line 238: Capital 't' for 'Today'.	Corrected	P10 line 257
	Page 13, line 289: For consistency, consider using the acronym for Health and Social Care Act as this was stated on the previous page (line 276-277).	Changed to HSCA	P12, L 296
	Page 18, line 443: "...in our study selected map closely..." – remove 'selected'.	Removed	P17 line 464
	References – it would be useful to include the year (and data source if applicable) within the reference description for the following references: 2, 6, 7, 9, 10, 13, 14, 15, 21, 22, 23, 24, and 27.	We have added dates for all these documents and websites.	P18-20

VERSION 2 – REVIEW

REVIEWER	Nadine Janis Pohontsch University Medical Center Hamburg-Eppendorf, Germany
REVIEW RETURNED	13-Jul-2020
GENERAL COMMENTS	Thank you for revising your article. All my comments were adequately addressed.